# Morphological Changes and Component Characterization of Coffee Silverskin

**DOI:** 10.3390/molecules26164914

**Published:** 2021-08-13

**Authors:** Xia Wang, De-Fu Hong, Gui-Lin Hu, Zhong-Rong Li, Xing-Rong Peng, Qiang-Qiang Shi, Ming-Hua Qiu

**Affiliations:** 1State Key Laboratory of Phytochemistry and Plant Resources in West China, Kunming Institute of Botany, Chinese Academy of Sciences, Kunming 650201, Yunnan, China; wangxia@mail.kib.ac.cn (X.W.); hongdefu@mail.kib.ac.cn (D.-F.H.); huguili@mail.kib.ac.cn (G.-L.H.); lizhongrong@mail.kib.ac.cn (Z.-R.L.); pengxingrong@mail.kib.ac.cn (X.-R.P.); shiqiangqiang@mail.kib.ac.cn (Q.-Q.S.); 2University of Chinese Academy of Sciences, Beijing 100049, China

**Keywords:** NMR spectroscopy, coffee silverskin, qualitative analysis, quantitative analysis

## Abstract

Nuclear magnetic resonance (NMR) spectroscopy was used for the qualitative and quantitative analysis of aqueous extracts of unroasted and roasted coffee silverskin (CS). Twenty compounds were identified from 1D and 2D NMR spectra, including caffeine, chlorogenic acid (CGA), trigonelline, fructose, glucose, sucrose, etc. For the first time, the presence of trigonelline was detected in CS. Results of the quantitative analysis showed that the total amount of the main components after roasting was reduced by 45.6% compared with values before roasting. Sugars in the water extracts were the main components in CS, and fructose was the most abundant sugar, its relative content accounting for 38.7% and 38.4% in unroasted and roasted CS, respectively. Moreover, 1D NMR combined with 2D NMR technology shows application prospects in the rapid, non-destructive detection of CS. In addition, it was observed by optical microscopy and scanning electron microscopy (SEM) that the morphology of CS changed obviously before and after roasting.

## 1. Introduction

According to the International Coffee Organization (ICO), the global production of green coffee beans reached 1.014 million tons in 2019 [1]. The processing of coffee cherries results in the generation of a large number of coffee by-products, including pulp, parchment, coffee silverskin (CS), and coffee grounds. The effective utilization of these coffee by-products is not only environmentally beneficial but also conducive to the sustainable development of the coffee industry. Therefore, in recent years, many studies have begun to pay attention to the application of the wastes from coffee bean processing, and these studies have been summarized [2,3]. Among them, CS has been proven to be rich in dietary fiber [4] and antioxidant substances [5,6], and it can be used as an additive for functional foods [7,8] and cosmetics [9], as well as in adsorbent materials [10,11], showing good product development potential.

CS is a thin layer of skin that adheres tightly to an oval coffee bean; it accounts for 4% of the total dry weight of a green coffee bean, accounting for 0.6% of the dry weight of coffee berries [12]. If mixed into coffee powder, it will increase the bitterness. CS is mainly produced in two stages during the production process: (a) during the shelling and peeling stage of green coffee beans before export, the CS is removed from the surfaces of coffee beans together with the seed shell, but it is not completely removed; (b) after roasting, the whole CS of coffee is peeled from the surfaces of the coffee beans. Approximately 40 substances including cellulose, hemicellulose, fructose, sucrose, glucose, caffeine, chlorogenic acid, vitamin E, 5-hydroxymethylfuranal, and fatty acid have been reported in CS. Among the total ingredients, sugars, caffeine, and chlorogenic acid have received special attention as functional substances in CS. Borrelli et al. [12] used enzyme gravity and chemical methods to determine the composition of CS and found that the main components were polysaccharides and proteins; total dietary fiber reached 60%, and soluble dietary fiber reached 8%. Ballesteros et al. [4] analyzed the cellulose, hemicellulose, lignin, and other sugar components’ content in CS by HPLC, and their results showed that water-insoluble cellulose and hemicellulose accounted for 40.45% of the total mass. Martinez-Saez et al. [13] used capillary electrophoresis to measure the caffeine and chlorogenic acid content in CS and found that their quantities were 3.02 g/100 g and 751.2 mg/100 g, respectively, while another study based on LC–MS found that their quantities were 1.0 g/100 g and 198.7 mg/100 g, respectively [14]. The difference between these measurement data may be due to the inherent differences in the CS itself, such as its species, source, and processing method, or the use of different analysis methods. It should be noted that the above analysis and identification methods for CS require complicated sample preparation processes, which may affect the results regarding the composition of CS.

NMR characterization is the use of a variety of NMR detection methods to directly obtain the complete composition information of complex samples. It is non-destructive, simple, and fast, and it can be used to perform qualitative and quantitative analysis on a variety of substances at the same time. It is now widely used in the food industry [15,16,17]. At the same time, NMR spectroscopy has been successfully used in the analysis of the composition of green coffee [18], roasted coffee [19], the effect of roasting on coffee composition [20], and coffee flavor [21]. In addition, our research team is proficient in the use of NMR to characterize the chemical composition of coffee [22,23,24]. However, there is no research using this method to characterize the composition of CS.

Thus far, the relative content of caffeine, chlorogenic acid, sugars, and other major bioactive components in green coffee or roasted coffee beans has been fully studied. However, the main chemical compositions of CS, especially unroasted CS, have not been clarified. Therefore, in order to obtain comprehensive information on the compositions of unroasted and roasted CS, we chose to use NMR spectroscopy to characterize a water extract of CS.

## 2. Results and Discussion

### 2.1. Changes in Microstructure of CS

It was found by light microscopy that the melanin produced by the reaction of the sugars and the amine compounds via the Maillard reaction in lightly roasted CS was unevenly distributed in the center of the cells (Figure 1B), indicating that the compounds involved in the reaction during roasting are unevenly distributed in CS cells. Comparing the scanning electron microscopy (SEM) photographs of deep roasted and unroasted CS, it was found that the CS before roasting exhibited a complete cell outline (Figure 1C), and the cell structure of the CS after deep roasting was destroyed, making it difficult to distinguish the contours of the cells (Figure 1D).

### 2.2. Qualitative Analysis of CS

In this study, ^1^H NMR was used to compare the composition of the heavy water extract of CS before and after roasting (Appendix A). The spectra showed that there was no significant difference in the composition of the CS before and after roasting. Twenty compounds were identified with the aid of 1D and 2D NMR spectra (Figure 2). In addition, the 1D NMR spectrum of lightly roasted A1 was used as an example to identify 20 compounds (Appendix A). The main compounds in the aqueous extract of CS were sugars, organic acids, alkaloids, and amino acids, including caffeine, trigonelline, 5-chlorogenic acid (5-CGA), fructose, sucrose, glucose, malic acid, and glycine.

#### 2.2.1. Alkaloids

As active alkaloids in coffee bean, caffeine and trigonelline are widely distributed in various parts of coffee, including the seeds, flowers, honey, and leaves [25,26,27]. However, previous analyses of the composition of CS have not detected the presence of trigonelline. The possible reasons for this are: (a) the roasting degree of the CS sample itself has a greater influence on the trigonelline content, and roasting causes the degradation of trigonelline; (b) the measurement method is for a certain kind of compound. As a result, researchers have not considered the existence of trigonelline during HPLC and LC–MS analysis, so the existence of trigonelline in CS has not been reported. To the best of our knowledge, this is the first time that the presence of trigonelline has been detected in CS. Meanwhile, trigonelline has been shown to exert multiple biological activities, including anti-diabetic [28], anti-obesity [29], anti-inflammatory [30], and neuroprotective effects [31]. Therefore, the discovery of trigonelline increases the potential of CS application.

#### 2.2.2. Sugars

During the signal assignment process, there was serious signal overlap between the 1D NMR data of the carbohydrate components, but six sugars, including *α*-d-glucopyranose, *β*-d-glucopyranose, *β*-d-pyranose, *β*-d-fructofuranose, sucrose, and inositol, were determined by the correlation signals of the C/H core in the same coupling system in the HSQC-TOCSY spectrum or other 2D NMR spectrum (^1^H-^1^H COSY, HSQC, and HMBC spectrum). Taking the 2D NMR spectrum of sample A1 as an example to analyze sucrose (Figure 3), starting from the correlation peaks of the terminal hydrogen *δ*_H_ 5.41 (H-1, *J* = 3.9 Hz) and the terminal carbon *δ*_C_ 92.1 (C-1) of the glucose in sucrose, the process of confirming signal attribution by 2D NMR is as follows: straight line from the low field to the high field along the ^13^C spectrum and the ^1^H spectrum. The carbon spectrum has *δ*_C_ 69.2, 72.5, 71.1, and the hydrogen spectrum has *δ*_H_ 3.84 and 3.54. Further, four groups of signals, *δ* 69.2 → 3.46 (C-4), *δ* 72.5 → 3.76 (C-3), *δ* 71.1 → 3.54 (C-2), *δ* 72.4 → 3.84 (C-5), were determined by the HSQC spectrum. In the 1H-1H COSY spectrum, *δ* 5.41 (H-1) → 3.54 (H-2), 3.54 → 3.76 (H-3), 3.76 → 3.46 (H-4), 3.46 → 3.84 (H-5). C-6 and C-1 had no correlation signals, but C-5 (*δ* 72.4, 3.84) had a TOCSY correlation with *δ* 3.82, 60.5, respectively, and *δ* 60.5→3.82 in the HSQC spectrum, thus indicating that *δ* 3.82, 60.5 (C-6) belongs to the same coupling system.

The structural determination of the fructose fraction of sucrose began with the HMBC association between *δ*_H_ 5.41 (H-1) and *δ*_C_ 103.6 (C-1′). *δ*_H_ 3.67 had HMBC-related signals with *δ*_C_ 103.6, which was determined as C-5′ in combination with the HSQC signal (*δ*_C_ 62.3). In the HSQC-TOCSY spectrum, starting from C-5′, along the direction of the hydrogen spectrum, there were *δ*_H_ 3.82, 3.88, 4.05, 4.21 from the high field to the low field. Four groups of signals, *δ* 3.82 → 60.1 (C-6′), *δ* 3.88 → 81.3 (C-4′), *δ* 4.05 → 74.0 (C-3′), *δ* 4.21 → 76.4 (C-2′), were obtained by combining the HSQC spectra and the connection sequence *δ*_H_ 4.21 (H-2′) → 4.05 (H-3′) → 3.88 (H-4′) was determined by ^1^H-^1^H COSY. Finally, the above signals were confirmed to be in the same coupling system by the relevant signals in the HSQC-TOCSY spectrum of C/H-2′. Based on the above analysis, it was determined that sucrose was contained in the CS. The main HSQC-TOCSY-related signals of other saccharide components are shown in Appendix A. The signal assignment processes for other compounds by the 1D and 2D NMR spectra were the same as those for sucrose.

#### 2.2.3. Phenolic Acids

The phenolic acids in CS are mainly composed of CGA, cinnamoylquinic acid, feruloylquinic acid, and their isomers, among which CGA is the most abundant. CGA in the coffee beans mainly includes 3-CGA, 4-CGA, and 5-CGA. Wei et al. [18] used the ^1^H NMR spectrum to determine the presence of these three isomers by the difference between the esterification positions of quinic acid and the coupling constant *J* between adjacent H. Using this method, the phenolic acid with the highest content in CS was determined to be 5-CGA. Gokhan Zengin et al. [32] used HPLC–MS/MS to analyze the active compounds of silver peel extracted by water, and they found that the content of 5-CGA was the highest, followed by 4-CGA. After careful analysis of 2D NMR, although some structural fragments of other acids can be found (Appendix A), their structures cannot be determined, which may be due to the small amount of CS samples extracted, resulting in no corresponding NMR signals. This reflects the comprehensive and non-destructive characteristics of NMR analysis information.

The 1D NMR data of the 20 compounds are listed in Table 1, and the signal assignments are detailed in Appendix A. The 1D NMR data of all compounds are consistent with the literature data [18,19,20,21]. Additionally, we collected the NMR data under the same conditions after mixing the standard materials, and the results were in agreement with the experimental data (Appendix A).

### 2.3. Quantitative Analysis of CS

TSP was used as an internal standard substance and the characteristic peak ^1^H-NMR signal of the CS components was compared with the ^1^H-NMR signal of TSP to obtain the relative content of CS components and the characteristic peak chemical shift, and a number of protons are shown in Appendix A. The relative integral areas of different compounds before (B) and after roasting (A) are shown in Figure 4A. Garcia De Serna et al. [33] used CS as a sucrose substitute and stevia to improve the formulation of biscuits. Meanwhile, the results showed that the sugars in the water extract of CS before or after roasting were the main component of the active components, accounting for 87.2% and 77.1%, respectively, and fructose was the most abundant among the saccharides, accounting for 49.6% and 44.5%, respectively. Therefore, the analysis results further confirm that CS is rich in carbohydrates and can be used for food development.

In contrast to coffee beans, the relative content of caffeine was significantly higher than that of 5-CGA in CS (Appendix A), which is consistent with the results obtained by Iriondo-DeHond et al. [34], who used UPLC–MS/MS to determine the content of caffeine and 5-CQA. At the same time, the lowest reported caffeine content in CS was 4.4 mg/g [35]. Therefore, it is presumed that CS contains a certain amount of trigonelline. It is worth noting that the content of trigonelline in CS, which has not been detected before, was higher than that of caffeine (Appendix A). The total amounts of the main ingredients after roasting were reduced by 45.6% compared with values before roasting, which was mainly due to the Maillard reaction between sugar components (Figure 4B). Caffeine did not change significantly as it remained thermally stable during the roasting process. Trigonelline and 5-CGA showed only a small amount of degradation due to the shallow roasting (Figure 4B).

## 3. Materials and Methods

### 3.1. Instruments and Materials

The CS samples selected in this study were obtained from *Arabica* coffee planted in Yunnan, China. A Paic JB-3 coffee roaster was used for sample roasting. The samples were ground with a Jiuyang JYL-B060; a Kz-2758 drying oven was used for sample moisture control. SEM images were acquired with a Sigma300 (CARL ZEISS) field emission scanning electron microscope. Optical micrograph images were acquired with a dark-field microscope (Zhejiang Sunny Optical Co., Ltd., Ningbo, China). D_2_O (99.9%) required for extraction was purchased from Beijing Yinuokai Technology Co. Ltd. An SZCL-4B intelligent magnetic heating stirrer was used for the sample heating, stirring, and extraction; a Genevac miVac vacuum centrifugal concentrator was used for sample extract centrifugation. The sample NMR data were acquired on a Bruker Avance III 800 MHz NMR spectrometer. TSP was purchased from Shanghai Source Ye Biotechnology Co. Ltd. Other standards required for the experiment were purchased from Aladdin Reagent (Shanghai, China) Co., Ltd.

### 3.2. Sample Preparation

For dried fruits (1.5 kg), the pulp and parchment were removed to obtain 760 g of green beans covered with CS. Then, the beans were divided into three portions (A.350 g, B.350 g, C.60 g). Parts A and C were roasted with a coffee roaster to obtain lightly roasted beans (LRB) and dark-roasted beans (DRB), respectively. Part B was used for the collection of unroasted CS samples. The roasting degree was mirrored by the color value of the beans (refer to Specialty Coffee Association of America color cards): LRB, 80–95; DRB, 40–55.

The water content of CS samples A and B was determined by the weight loss method. The two samples were ground in a grinder, placed in a beaker, dried in a 50 °C oven at constant temperature, and weighed once every half hour until the difference between the two drying times was less than 0.3 g, so that the water content of the two samples was consistent, and they were sealed for storage. We accurately weighed the CS powder of 0.5 g × 2 A sample and B sample and placed them in 4 round-bottomed flasks (5 mL), respectively. We then sealed each sample and stirred it with 3.5 mL D_2_O at 60 °C (1500 r/m) for 2 h to obtain 4 sample water extracts. The resulting extracts were simply referred to as A1, A2, B1, and B2. After the sample was cooled at room temperature, the four extracts were transferred to PE tubes and centrifuged at 4000 rpm for 30 min; 400 μL supernatant and 10 μL TSP were separately added to the NMR tube; then, ^1^H-NMR and 2D NMR data of the D_2_O extracts of unroasted and lightly roasted CS samples were acquired using an 800 MHz NMR spectrometer.

### 3.3. NMR Data Acquisition Conditions

All 1D and 2D NMR data were collected on a Bruker Avance III 800 MHz NMR spectrometer at a temperature of 27 °C. All experimental parameters are listed in Appendix A.

### 3.4. Morphological Characterization of CS with Different Roasting Degrees

For light microscope image collection, unroasted (sample B) and lightly roasted (sample A) CS with similar shapes were selected, respectively. The surface was evenly immersed in oil, and then the sample was placed under dark-field microscopy for observation. For SEM characterization, unroasted (sample B) and deep-roasted (sample C) CS with similar shapes were selected. The surface was evenly plated with silver, and then placed under SEM (CARL ZEISS) to collect images. SEM magnification was 40 to 600 times.

### 3.5. Data Processing

All 1D and 2D NMR spectral data were processed using the MestReNova software (Mestrelab Research, Santiago de Compostela, Spain). For qualitative analysis, the components of CS water extracts A (roasted) and B (unroasted) were analyzed and identified by using the 1D NMR and 2D NMR spectra of related components [19,20,21,22], and the accuracy of component identification was confirmed by the NMR spectra of the mixed standard samples under the same conditions. For quantitative analysis, the characteristic ^1^H signals (Appendix A) of the main components in the ^1^H NMR spectrum were normalized and integrated with respect to the ^1^H signal of the internal standard TSP, and the characteristic peak areas As and Bs before and after roasting were obtained, respectively. Then, the relative content changes in each substance before and after roasting were calculated.

Calculation formula:
Relativecontent(%)=AxTs×100%Rateofchange(%)=(As−Bs)Bs×100%

Formula: Ax, the characteristic peak H signal area of CS component; Ts, the total area of the characteristic peak H signal of the CS components; As, the characteristic peak H signal area of roasted CS component; Bs, the characteristic peak H signal area of unroasted CS component.

## 4. Conclusions

The morphology of CS before and after roasting was observed by optical microscopy and SEM. The Maillard reaction occurred unevenly in the CS cells under light roasting, which may have resulted from the uneven distribution of sugars and amines in CS cells. Deeply roasted CS cells were severely damaged, making it difficult to distinguish their contours under SEM.

NMR spectroscopy is the most important tool to identify the structure of organic compounds, because it can provide a variety of 1D and 2D NMR spectral data, reflecting a large amount of structural information. Therefore, it is becoming increasingly popular for researchers to characterize mixed-component samples by NMR spectroscopy. The water extract of CS before and after roasting was characterized by NMR spectroscopy, and 20 compounds, including caffeine, 5-CGA, trigonelline, fructose, glucose, sucrose, etc., were identified. Meanwhile, trigonelline was first found in the water extract of CS, which increases the potential application prospects of CS. NMR spectroscopy was proven to be a convenient means to characterize CS compounds.

The relative content of the main components in CS was detected and their changes during the roasting process were discussed. Even though the Maillard reaction during the roasting process caused a large reduction in sugars, they were still the main ingredients in the aqueous extract of both unroasted and roasted CS. Moreover, 5-CGA and trigonelline underwent light degradation. In total, the main ingredients in roasted CS were reduced by 45.6% compared with unroasted CS. Considering that the utilization of CS is mainly based on its rich sugars, caffeine, CGA, and trigonelline content, the unroasted CS is more valuable for recycling than roasted CS. 

## Figures and Tables

**Figure 1 molecules-26-04914-f001:**
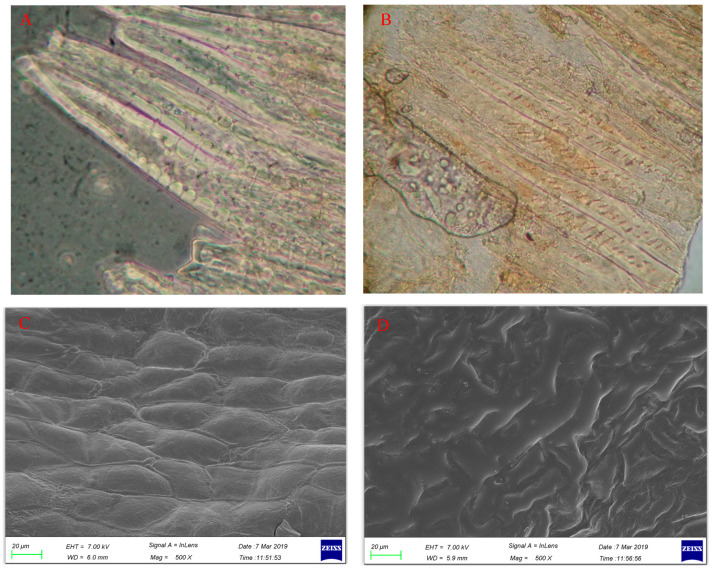
Microstructure of CS. (**A**): optical micrograph of unroasted CS (×1000); (**B**): optical micrograph of lightly roasted CS (×1000); (**C**): SEM graph of unroasted CS (×500); (**D**): SEM graph of deep-roasted CS (×500).

**Figure 2 molecules-26-04914-f002:**
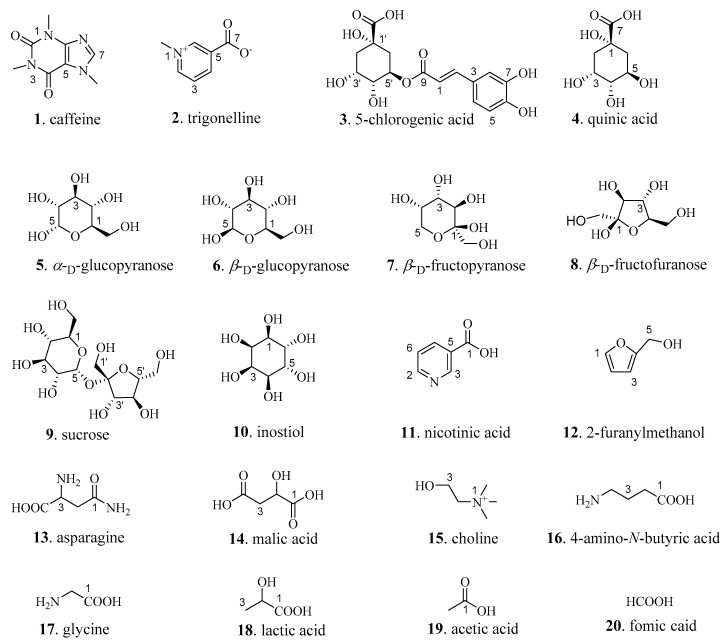
The compounds (**1**-**20**) of CS aqueous extract were identified by 2D NMR spectra.

**Figure 3 molecules-26-04914-f003:**
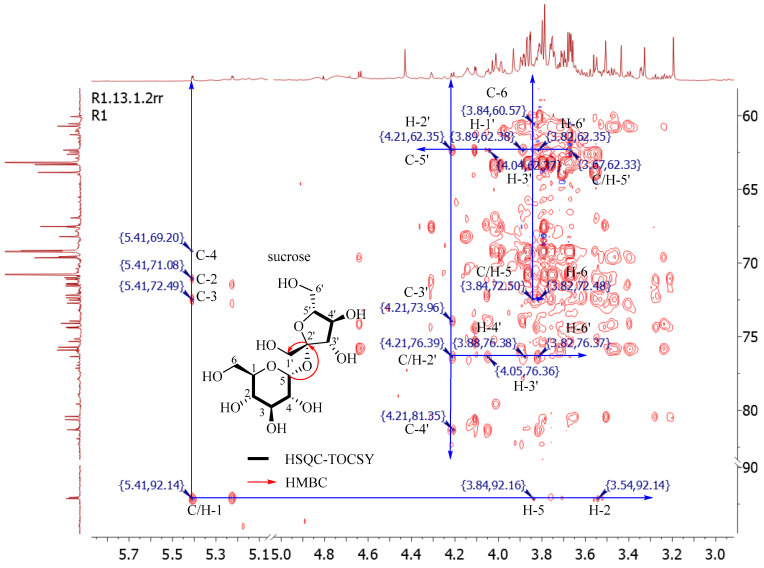
The HSQC-TOCSY correlations of sucrose from CS aqueous extract.

**Figure 4 molecules-26-04914-f004:**
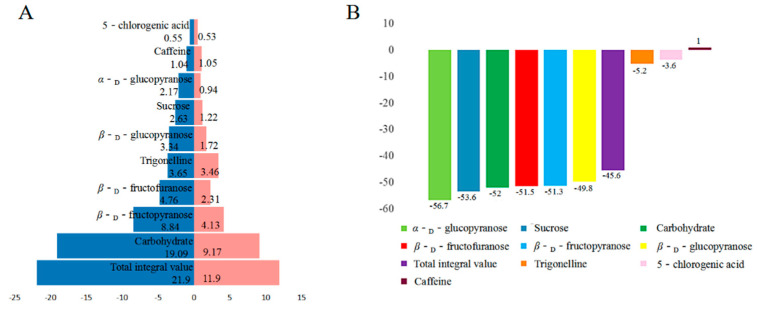
(**A**) the relative integral area of different compounds before (blue) and after roasting (pink); (**B**) the change rate of different compounds after roasting.

**Table 1 molecules-26-04914-t001:** The 1D NMR data of CS components.

Compound	^1^H	^13^C	Assignment	Compound	^1^H	^13^C	Assignment
Caffeine	3.33	27.9 (q)	N1-CH_3_	5-CQA	6.37	114.6 (d)	C1-H
-	-	152.7 (s)	C2-O	-	7.61	145.9 (d)	C2-H
-	3.51	29.8 (q)	N3-CH_3_	-	-	126.8 (s)	C3
-	-	156.4 (s)	C4-O	-	7.08	122.5 (d)	C4-H
-	-	107.9 (s)	C5	-	6.91	116.1 (d)	C5-H
-	3.93	33.4 (q)	N6-CH_3_	-	-	147 (s)	C6
-	7.87	143.5 (d)	C7-H	-	-	144.2 (s)	C7
-	-	148.4 (s)	C8	-	7.14	114.9 (d)	C8-H
Trigonelline	4.43	48.2 (q)	N1-CH_3_	-	-	169 (s)	C9
-	8.84	145.8 (d)	C2-H	-	5.32	71.1 (d)	C5′-H
-	8.07	127.5 (d)	C3-H	-	3.88	72.8 (d)	C4′-H
-	8.83	144.7 (d)	C4-H	-	4.26	70.6 (d)	C3′-H
-	-	136.9 (s)	C5	-	2.15, 2.05	37.3 (t)	C2′-H_2_
-	9.12	145.9 (d)	C6-H	-	-	77.3 (s)	C1′
-	-	167.8 (s)	C7	-	2.69	39.8 (t)	C6′-H_2_
					2.51		
Quinic acid	-	75.8 (s)	C1	-	-	173.6 (s)	C7′
-	2.05,	37.3 (t)	C2-H_2_	*α*-d-glucopyranose	3.76	72.5 (d)	C1-H
	1.98						
-	4.15	70.4 (d)	C3-H	-	3.46	69.2 (d)	C2-H
-	3.55	75.2 (d)	C4-H	-	3.71	72.7 (d)	C3-H
-	4.02	66.9 (d)	C5-H	-	3.53	71.5 (d)	C4-H
-	2.07, 1.88	40.6 (t)	C6-H_2_	-	5.23	92.07 (d)	C5-H
-	-	173.3 (s)	C7	-	3.67, 3.81	62.3 (t)	C6-H_2_
*β*-d-glucopyranose	3.83	71.4 (d)	C1-H	*β*-d-fructopyranose	-	98 (s)	C1
-	3.4	69.6 (d)	C2-H	-	3.79	67.5 (d)	C2-H
-	3.47	75.9 (d)	C3-H	-	3.89	69.6 (d)	C3-H
-	3.24	74.1 (d)	C4-H	-	3.99	69.2 (d)	C4-H
-	4.64	95.9 (d)	C5-H	-	4.02, 3.67	63.3 (t)	C5-H_2_
-	3.89, 3.71	60.7 (t)	C6-H_2_	-	3.55, 3.70	63.8 (t)	C6-H_2_
*β*-d-fructofuranose	-	101.5 (s)	C1	Sucrose	3.84	72.4 (d)	C5-H
-	4.11	74.4 (s)	C2-H	-	3.46	69.2 (d)	C4-H
-	4.11	75.4 (s)	C3-H	-	3.76	72.5 (d)	C3-H
-	3.82	80.6 (s)	C4-H	-	3.54	71.1 (d)	C2-H
-	3.67, 3.81	62.3 (t)	C5-H_2_	-	5.41	92.12 (d)	C1-H
-	3.55, 3.58	62.6 (t)	C6-H_2_	-	3.82	60.5 (t)	C6-H_2_
Inostiol	3.62	72.2 (d)	C1-H	-	3.67	62.3 (t)	C5′-H_2_
-	3.51	71.1 (d)	C2-H	-	-	103.6 (s)	C1′
-	4.31	71.3 (d)	C3-H	-	4.21	76.4 (d)	C2′-H
-	3.21	80.4 (d)	C4-H	-	4.05	74 (d)	C3′-H
-	3.65	71.6 (d)	C5-H	-	3.88	81.3 (d)	C4′-H
-	3.28	74.4 (d)	C6-H	-	3.82	60.1 (t)	C6′-H_2_
Nicotinic acid	-	166.2 (s)	C1	2-Furanylmethanol	7.52	146.2 (d)	C1-H
-	8.83	153.2 (d)	C2-H	-	6.67	111 (d)	C2-H
-	9.13	150.2 (d)	C3-H	-	6.65	110.8 (d)	C3-H
-	8.32	136.9 (d)	C4-H	-	-	161.4 (s)	C4
-	-	126.7 (s)	C5	-	4.68	56.0 (t)	C5-H_2_
-	7.50	123.71 (d)	C6-h	Malic acid	-	179.4 (s)	C1
Asparagine	-	174.4 (s)	C1	-	4.01	66.9 (d)	C2-H
-	2.86, 2.95	34.4 (t)	C2-H_2_	-	2.69, 2.51	38.3 (t)	C3-H_2_
-	4.0	51.2 (d)	C3-H	-	-	179.2 (s)	C4
-	-	173 (s)	C4	4-Amino-*N*-butyric acid	-	180.5 (s)	C1
Choline	3.19	53.8 (q)	N1-CH_3_	-	2.35	33.4 (t)	C2-H_2_
-	3.19	53.8 (q)	N1-CH_3_	-	1.91	23.1 (t)	C3-H_2_
-	3.19	53.8 (q)	N1-CH_3_	-	3.02	39.1 (t)	C4-H_2_
-	3.51	67.4 (t)	C2-H_2_	Lactic acid	-	182 (s)	C1
-	4.06	55.5 (t)	C3-H_2_	-	4.15	68.2 (d)	C2-H
Glycine	-	170.7 (s)	C1	-	1.34	19.9 (q)	C3-H_3_
-	3.81	52.8 (t)	C2-H_2_	Acetic acid	-	178.8 (s)	C1
Formic acid	8.43	171.4 (d)	C1-H	-	2.0	21.6 (d)	C2-H

## Data Availability

The 1D NMR data of the 20 compounds provided in this study are available in the article [18,19,20,21] and Appendix A.

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
