# Peer review of "Morphological Changes and Component Characterization of Coffee Silverskin"

_molecules, 2021, doi:10.3390/molecules26164914_

Round 1
Reviewer 1 Report
Wang et al. report on a NMR characterization of aqueous extracts of roasted and unroasted coffee silverskin. 1D and 2D spectra allowed the identification of 20 compounds, including sugars, phenolic acids and alkaloids. I acknowledge the relevance of a deep characterization of the material, aiming also at increasing its recycle. I have, however, major concerns on the manuscript, from the methodology and data analysis to the clarity of the discussion and appropriate language and style. Also, the supplementary
material is totally missing. I therefore recommend the paper to be rejected and I invite the authors to - at least partially - rewrite the manuscript following the comments below prior to another submission.
1. It should be “NMR spectroscopy” instead of “NMR technology”
2. The topic is not put adequately in the context and the state-of-the-art literature is not cited. From a quick literature search I could find many recent works on the subject, reporting for instance on the application of CS (10.3390/molecules26092676; 10.1016/j.wasman.2020.12.018; 10.3390/toxins13020132; 10.1007/s13197-021-05032-5; 10.1007/s10924-021-02067-9; 10.1016/j.foodres.2020.109733), its recovery (10.1016/j.fuproc.2020.106708; 10.1007/s12649-021-01435-9) and even its characterization (10.1016/j.foodres.2020.109128; 10.3390/molecules25132993; 10.3390/foods9060713).
3. Some sentences are not understandable in my opinion (i.e. p. 1 lines 35-37), and the experimental procedure is poorly described (all section 3), it is not a recipe (i.e. p. 8 lines 181-182) and you need full sentences. No detail is given for the NMR spectra in section 3.3 (type of experiment and acquisition parameters) and I do not really know what you mean with “nucleus tube” (p.9 line 191).
4. The supporting tables and graphics are missing, 1D and 2D spectra should be provided with proper assignment. Also, it is not clear which experiment was used from the main text, i.e. p.4 line 95 “other 2D NMR spectra”.
5. Numbering for compounds in Fig. 2 is needed to understand the assignment described in the text and reported in Table 1.
6. The methodology used for data analysis is not clear. Were the integrals normalized according to the number of protons? For a proper quantitative analysis, the absolute quantification is typically performed, using the known amount of the reference compound (TSP), why did you report “relative integrals”? And how exactly did you calculate the negative and positive relative integrals depicted in Fig 4A?
Author Response
Dear Reviewers:
Our manuscript entitled “Morphological Changes and Components Characterization of Coffee Silverskin” was used as a special issue "A Themed Issue in Memory of Academician Xing Qiyi (1911–2002)". At the same time, thank you for the reviewers’ comments concerning our manuscript entitled “Paper Title” (ID: molecules-1292519). Those comments are all valuable and very helpful for revising and improving our paper, as well as the important guiding significance to our researches. We have studied comments carefully and have made correction which we hope meet with approval. Revised portion are marked in red in the paper. The main corrections in the paper and the responds to the reviewer’s comments are as flowing:
Point 1: It should be “NMR spectroscopy” instead of “NMR technology”. 

Response 1: “NMR technology” has been replaced by “NMR spectroscopy” in the manuscript.
Point 2: The topic is not put adequately in the context and the state-of-the-art literature is not cited. From a quick literature search I could find many recent works on the subject, reporting for instance on the application of CS (10.3390/molecules26092676; 10.1016/j.wasman.2020.12.018; 10.3390/toxins13020132; 10.1007/s13197-021-05032-5; 10.1007/s10924-021-02067-9; 10.1016/j.foodres.2020.109733), its recovery (10.1016/j.fuproc.2020.106708; 10.1007/s12649-021-01435-9) and even its characterization (10.1016/j.foodres.2020.109128; 10.3390/molecules25132993; 10.3390/foods9060713).
Response 2: After reading the article you provided, the relevant article has been cited in the manuscript.
Point 3: Some sentences are not understandable in my opinion (i.e. p. 1 lines 35-37), and the experimental procedure is poorly described (all section 3), it is not a recipe (i.e. p. 8 lines 181-182) and you need full sentences. No detail is given for the NMR spectra in section 3.3 (type of experiment and acquisition parameters) and I do not really know what you mean with “nucleus tube” (p.9 line 191).
Response 3: a). The sentence in lines 35-37 on page 1 was modified by me to read: “CS is a thin layer of skin that adheres tightly to an oval coffee bean, which accounts for 4% of the total dry weight of green coffee bean, accounting for 0.6% of the dry weight of coffee berries”;
b). I re-completed the description of the experiment process.
c). The experiment types and acquisition parameters in section 3.3 have been added to Table S1 by me.
d). "nucleus tube" has been revised to “NMR tube”.
Point 4: The supporting tables and graphics are missing, 1D and 2D spectra should be provided with proper assignment. Also, it is not clear which experiment was used from the main text, i.e. p.4 line 95 “other 2D NMR spectra”.
Response 4: a.) In the support material, we took the A1 sample (lightly baked CS) as an example, compared it with literature data records and standard mixtures, and completed the characterization of 20 compounds. At the same time, according to the NMR data of the compounds in table1, the 2DNMR assignment of sugars was completed.
b.) Other 2D NMR spectra refer to 1H-1H COSY, HSQC, and HMBC spectra, and have been added to the manuscript.
Point 5: Numbering for compounds in Fig. 2 is needed to understand the assignment described in the text and reported in Table 1.
Response 5: 20 compounds have been numbered and displayed in Fig. 2.
Point 6: The methodology used for data analysis is not clear. Were the integrals normalized according to the number of protons? For a proper quantitative analysis, the absolute quantification is typically performed, using the known amount of the reference compound (TSP), why did you report “relative integrals”? And how exactly did you calculate the negative and positive relative integrals depicted in Fig 4A?
Response 6: a.) We use the H signal of the internal standard TSP as the standard for integration.
b.)This research mainly studies the changes in the chemical composition of the silver skin before and after roasted. Therefore, the relative integration is used to discuss the changes in the chemical composition.
c.)The calculation formula has been added to the manuscript.
Reviewer 2 Report
The work of Wang, Hong, et al. is a useful contribution to the quantitative chemical analysis of the coffee silverskin (CS) before and after roasting. This chemical analysis is performed by using standard 1D and 2D NMR techniques (not technology) to unravel complex NMR spectra with overlapping peaks and characterize the numerous compounds in CS before and after roasting. The authors clearly show that roasting of CS decreases the quantity of valuable CS ingredients by almost 50 % so that these ingredients should be recovered from unroasted CS considered as waste from coffee production. They also discovered the presence in CS of trigonelline, a plant alkaloid with many therapeutic activities. This work deserves publication in Molecules after the authors have clarified the following issues.
Critical issues.
As I understand it, the goal of drying in subsection 3.2 is to prepare two really dry samples of equal weights of unroasted and roasted CS, and then to extract the valuable compounds from both samples with the same volume of hot heavy water in order to properly compare the amounts of organic reactants in unroasted CS with those which remain in the roasted state after the Maillard reaction due to roasting.
Does the drying of the large portions of unroasted and roasted CS to within 0.3 g mean that full drying has been reached for both portions? The critical drying procedure should be unambiguously explained.
Deep and slight roasting processes should be defined in terms of roasting temperature and duration.
Minor issues.
In the introduction, does the annual coffee production of about 9.4 million tons correspond to fresh or dry product?
What is the annual production of dry CS in tons per year?
In Fig. 4A, the relative quantities of valuable compounds extracted from unroasted CS are positive numbers and could be represented in this way simply by using two opposite half-axes with positive graduations.
The term "NMR tube" should be used instead of "nucleus tube".
The authors could discuss the economic interest of recovering ingredients from unroasted and roasted CS depending on the technical difficulties and cost of working with these materials.
They could also evaluate the relative importance of Arabica coffee seeds and CS as complementary sources of trigonelline.
Author Response
Dear Reviewers:
Thank you for your letter and for the reviewers’ comments concerning our manuscript entitled “Paper Title” (ID: molecules-1292519). Those comments are all valuable and very helpful for revising and improving our paper, as well as the important guiding significance to our researches. We have studied comments carefully and have made corrections which we hope meet with approval. Revised portions are marked in red on the paper. The main corrections in the paper and the responses to the reviewer’s comments are as flowing:
Point 1: Does the drying of the large portions of unroasted and roasted CS to within 0.3 g mean that full drying has been reached for both portions? The critical drying procedure should be unambiguously explained.

Response 1: The two samples have not reached a completely dry state. This step of removing water is to ensure that the water content of the two samples remains the same. The drying program has been modified and described as: “We use the weighloss method to determinet the moisture content of silver skin samples, the two parts of the samples were ground in a grinder, placed in a beaker, placed in a drying oven at 50 oC and dried at a constant temperature, and weigh it every half an hour until the difference between the two dryings is within 0.3 g”.
Point 2: In the introduction, does the annual coffee production of about 9.4 million tons correspond to fresh or dry product?What is the annual production of dry CS in tons per year?
Response 2: About 1.014million tons of coffee in 2019 corresponds to green coffee beansï¼›In order to better understand you and the readers, this part has been modified to: “According to the International Coffee Organization (ICO), global coffee production of green coffee beans reached 1.014 million tons in 2019.”
The annual output of dry CS can not be counted, because the silver husk is mainly produced in two stages in the production process, one is the shelling and peeling stage before the export of green coffee beans; Second, the intact coffee silver peel falls off the surface of coffee beans after baking, but its quality accounts for 4% of the total dry coffee beans, which is estimated to be 400,000 tons.
Point 3: In Fig. 4A, the relative quantities of valuable compounds extracted from unroasted CS are positive numbers and could be represented in this way simply by using two opposite half-axes with positive graduations.
Response 3: Figure 4A has been modified by your suggestion.
Point 4: The term "NMR tube" should be used instead of "nucleus tube".
Response 4: "nucleus tube" has been revised to “NMR tube”.
Point 5: The authors could discuss the economic interest of recovering ingredients from unroasted and roasted CS depending on the technical difficulties and cost of working with these materials.
Response 5: I'm very sorry. Consulting a large number of documents mainly focuses on the use of CS bioactive compounds, and there is no economic discussion on the technical difficulty and cost. However, in the manuscript, we made a similar discussion. According to the changes in the active material before and after roasting, it is suggested that the unfired CS is more valuable than the roasting CS.
Reviewer 3 Report
The submitted manuscript presents the results of liquid NMR analysis of coffee silverskin. Though the Authors have done some work with acquiring the spectra and performing peaks assignment, the work needs major revision.
First, the “results and discussion” part is merely presentation and description of the results. There is no real discussion. The Authors should compare the NMR method with previously reported analytical studies on CS analysis. I.e. they should compare the quantitative results, pros and cons of each method etc.,
Second, in the Materials&Methods section there is no information about the number of samples that have been analyzed using ssNMR.
Finally, in the conclusions it should be stated if designed method can be used as routine in the CS analysis.
Line 14, instead of „technology” it should be “spectroscopy”
Line 15, it should be separated into two sentences
Line 27, why do you present data reported 5 years ago?
Line 27, “production has produced”-it doesn’t sound good
Line 50, I wouldn’t put all the blame on different methods. Those differences are probably due to the different origin of the studied materials.
Line 55, what do you mean by “non-destructive”? After all, you have to add some deuterated solvent to the sample which makes the sample not appropriate for other analysis after NMR. So, please prove that NMR is non-destructive.
Line 57, instead of „technology” it should be “spectroscopy”
In the introduction, please state if the CS has been studied using NMR previously or not.
Lines 176 and 194-the information is duplicated
Table S1 is not available, you have forgotten to upload the supplementary materials.
Line 80, it should be “spectra” not “spectrum”
Lines 121-122, why those acids are not listed in Figure 2?
Figure 4, there is no such compound as B-D-pyranose.
Figure 4C is not very informative, please move it to SI.
Line 214, why? What are the benefits of trigonelline?
Line 230, the part on each author’s individual contribution is missing.
Author Response
Dear Reviewers:
Thank you for your letter and for the reviewers’ comments concerning our manuscript entitled “Paper Title” (ID: molecules-1292519). Those comments are all valuable and very helpful for revising and improving our paper, as well as the important guiding significance to our researches. We have studied comments carefully and have made corrections which we hope meet with approval. Revised portions are marked in red on the paper. The main corrections in the paper and the responses to the reviewer’s comments are as flowing:
Point 1: The “results and discussion” part is merely presentation and description of the results. There is no real discussion. The Authors should compare the NMR method with previously reported analytical studies on CS analysis. I.e. they should compare the quantitative results, pros and cons of each method etc.,
Response 1: The results and discussion parts have been modified by me. In addition, this study mainly discusses the changes in the content of active ingredients before and after roasting. The active ingredients were not quantitatively analyzed. Although the experimental results cannot be compared with other quantitative methods, we have added conduct component content analogy analysis. E.g: “Contrary to coffee beans, the relative content of caffeine is significantly higher than that of 5-CGA in CS (Figure S.14), which is consistent with the results obtained by Iriondo-DeHond et al. using UPLC-MS/MS to determine the content of caffeine and 5-CQA. At the same time, the lowest reported caffeine content in coffee silver peel is 4.4 mg/g, so it is speculated that there is a certain amount of trigonelline in CS”.
Point 2: In the Materials&Methods section there is no information about the number of samples that have been analyzed using NMR.
Response 2: In the Materials & Methods section, the information on the number of samples analyzed by NMR has been described in the manuscript.
Point 3: In the conclusions it should be stated if designed method can be used as routine in the CS analysis.
Response 3: NMR spectrum can be used as a routine for CS analysis and I have added it to the conclusion I added as follows:“Through the results of our research, NMR spectroscopy can be used as a conventional means for characterizing CS compounds”.
Point 4: Line 14, instead of “technology” it should be “spectroscopy”
Response 4: “NMR technology” has been replaced by “NMR spectroscopy” in the manuscript.
Point 5: Line 15, it should be separated into two sentences
Response 5: Line 15 has been modified to read as follows: “Nuclear Magnetic Resonance (NMR) technology was used for the qualitative and quantitative analysis of aqueous extracts of unroasted and roasted coffee silverskin (CS). 20 compounds were identified from 1D and 2D NMR spectra, including caffeine, chlorogenic acid (CGA), trigonelline, fructose, glucose, sucrose, etc.”
Point 6: Line 27, why do you present data reported 5 years ago?
Response 6: The data from 5 years ago has been revised to the coffee data for 2019 (According to the International Coffee Organization (ICO), global coffee production reached 1.014 million tons in 2019.)
Point 7: Line 27, “production has produced”-it doesn’t sound good
Response 7: The “production has produced” has been modified to“the processing of coffee cherries results in the generation of a large amount of coffee by-products”.
Point 8: Line 50, I wouldn’t put all the blame on different methods. Those differences are probably due to the different origin of the studied materials.
Response 8: According to your suggestion, this part was modified to “The difference between these measurement data may depend on the inherent difference of the silver skin itself, such as their species, source and processing method, or different analysis methods”.
Point 9: Line 55, what do you mean by “non-destructive”? After all, you have to add some deuterated solvent to the sample which makes the sample not appropriate for other analysis after NMR. So, please prove that NMR is non-destructive.
Response 9: HPLC and LC-MS are difficult to recover samples for analysis, and LC-MS can damage the composition of the samples. However, NMR collected the H spectrum of the sample and did not damage the sample; in addition, after the deuterated reagent is dried, the next step can be performed for other analysis.
Point 10: In the introduction, please state if the CS has been studied using NMR previously or not.
Response 10: I wrote in the introduction as follows: “However, there is no research using this method to characterize the composition of coffee silver skin.”
Point 11: Lines 176 and 194-the information is duplicated
Response 11: I carefully checked Lines 176 and 194 and found no duplicate information.
Point 12: Table S1 is not available, you have forgotten to upload the supplementary materials.
Response 12: Table S1 has been uploaded to the supplementary material.
Point 13: Line 80, it should be “spectra” not “spectrum”
Response 13: “spectrum” has been replaced by “spectra” in the manuscript.
Point 14: Lines 121-122, why those acids are not listed in Figure 2?
Response 14: Because this method determines that the most abundant content in CS peel is 5-chlorogenic acid, other chlorogenic acid components only get some structural fragments due to the small amount of extraction, so they are not listed in Figure 2.
Point 15: Figure 4, there is no such compound as β-D-pyranose.
Response 15: “β-D-pyranose” has been corrected to “β-D-fructopyranose”, as shown in Figure 4
Point 16: Figure 4C is not very informative, please move it to SI.
Response 16: Fig. 4C was moved to SI.
Point 17: Line 214, why? What are the benefits of trigonelline?
Response 17: “Trigonelline has been shown to have multiple biological activities including anti-diabetic, anti-obesity, anti-inflammatory and neuroprotective effects. Therefore, the discovery of trigonelline increases the possibility of CS application.”
Point 18: Line 230, the part on each author’s individual contribution is missing.
Response 18: The author’s contribution was supplemented by me as follows: “Wang Xia completed the experimental part of the article; De-fu Hong and Gui-lin Hu completed experimental and wrote the manuscript; Zhong-Rong Li, Xing-Rong Peng, Qiang-Qiang Shi completed in the experimental process guidance. Minghua Qiu contributed to the entire experiment design, supervision and management.”
Reviewer 4 Report
Dear Authors,
the paper “Characterization of Coffee Silverskin (CS) Components Using NMR Technology” is an interesting study with the intent to support the resource-use efficiency of coffee by products. The study obviously followed a well-planned and structured design.
Overall the manuscript was well written, and showed to some extend that the roasting process has an effect on CS components. Before publication, however, the manuscript needs revision on some major aspects.
The title only mentions the application of NMR Technology, but the study includes more techniques (SEM, optical microscopy). With the current title, readers might assume that this is a NMR methodology paper. But the focus of the manuscript is on the characterization of CS and not on the NMR methodology. The title must reflect that approach.
The authors also cite some papers about application of NMR to coffee, but they have not included or considered most recent developments in the field. The authors should therefore consider and include most recent publications.
A very comprehensive review on the Coffe By-products including CS was provided by Klingel et al. in 2020 (Klingel, T.; Kremer, J.I.; Gottstein, V.; Rajcic de Rezende, T.; Schwarz, S.; Lachenmeier, D.W. A Review of Coffee By-Products Including Leaf, Flower, Cherry, Husk, Silver Skin, and Spent Grounds as Novel Foods within the European Union. Foods 2020, 9, 665. https://doi.org/10.3390/foods9050665). While this review shows that CS has already been well characterized by other authors, the study “Characterization of Coffee Silverskin (CS) Components Using NMR Technology” provides new insight in the composition of CS before and after roasting. The review also includes recent studies with relevance to this paper. Therefore the authors should consider to include these findings and papers and discuss them in light of their own results.
The authors state that the discovery of trigonelline increases the possibility of CS application. This claim should be further elaborated and described in the paper.
The authors provided only dearth information on the data acquisition conditions of the NMR spectra. Transparency and traceability of sciences requires the authors to provide more information on the data acquisition, the applied methods and instrument parameters.
Author Response
Dear Reviewers:
Our manuscript entitled “Morphological Changes and Components Characterization of Coffee Silverskin” was used as a special issue "A Themed Issue in Memory of Academician Xing Qiyi (1911–2002)". At the same time, thank you for the reviewers’ comments concerning our manuscript entitled “Paper Title” (ID: molecules-1292519). Those comments are all valuable and very helpful for revising and improving our paper, as well as the important guiding significance to our researches. We have studied comments carefully and have made corrections which we hope meet with approval. Revised portions are marked in red on the paper. The main corrections in the paper and the responses to the reviewer’s comments are as flowing:
Point 1: It should be “NMR spectroscopy” instead of “NMR technology”. 

Response 1: “NMR technology” has been replaced by “NMR spectroscopy” in the manuscript.
Point 2: The topic is not put adequately in the context and the state-of-the-art literature is not cited. From a quick literature search I could find many recent works on the subject, reporting for instance on the application of CS (10.3390/molecules26092676; 10.1016/j.wasman.2020.12.018; 10.3390/toxins13020132; 10.1007/s13197-021-05032-5; 10.1007/s10924-021-02067-9; 10.1016/j.foodres.2020.109733), its recovery (10.1016/j.fuproc.2020.106708; 10.1007/s12649-021-01435-9) and even its characterization (10.1016/j.foodres.2020.109128; 10.3390/molecules25132993; 10.3390/foods9060713).
Response 2: After reading the article you provided, the relevant article has been cited in the manuscript.
Point 3: Some sentences are not understandable in my opinion (i.e. p. 1 lines 35-37), and the experimental procedure is poorly described (all section 3), it is not a recipe (i.e. p. 8 lines 181-182) and you need full sentences. No detail is given for the NMR spectra in section 3.3 (type of experiment and acquisition parameters) and I do not really know what you mean with “nucleus tube” (p.9 line 191).
Response 3: a). The sentence in lines 35-37 on page 1 was modified by me to read: “CS is a thin layer of skin that adheres tightly to an oval coffee bean, which accounts for 4% of the total dry weight of green coffee bean, accounting for 0.6% of the dry weight of coffee berries”;
b). I re-completed the description of the experiment process.
c). The experiment types and acquisition parameters in section 3.3 have been added to Table S1 by me.
d). "nucleus tube" has been revised to “NMR tube”.
Point 4: The supporting tables and graphics are missing, 1D and 2D spectra should be provided with proper assignment. Also, it is not clear which experiment was used from the main text, i.e. p.4 line 95 “other 2D NMR spectra”.
Response 4: a.) In the support material, we took the A1 sample (lightly baked CS) as an example, compared it with literature data records and standard mixtures, and completed the characterization of 20 compounds. At the same time, according to the NMR data of the compounds in table1, the 2DNMR assignment of sugars was completed.
b.) Other 2D NMR spectra refer to 1H-1H COSY, HSQC, and HMBC spectra, and have been added to the manuscript.
Point 5: Numbering for compounds in Fig. 2 is needed to understand the assignment described in the text and reported in Table 1.
Response 5: 20 compounds have been numbered and displayed in Fig. 2.
Point 6: The methodology used for data analysis is not clear. Were the integrals normalized according to the number of protons? For a proper quantitative analysis, the absolute quantification is typically performed, using the known amount of the reference compound (TSP), why did you report “relative integrals”? And how exactly did you calculate the negative and positive relative integrals depicted in Fig 4A?
Response 6: a.) We use the H signal of the internal standard TSP as the standard for integration.
b.)This research mainly studies the changes in the chemical composition of the silver skin before and after roasted. Therefore, the relative integration is used to discuss the changes in the chemical composition.
c.)The calculation formula has been added to the manuscript.
Round 2
Reviewer 1 Report
Please see the attachment.

Author Response
Dear Reviewers:
Our manuscript entitled “Morphological Changes and Components Characterization of Coffee Silverskin” was used as a special issue "A Themed Issue in Memory of Academician Xing Qiyi (1911–2002)". At the same time, thank you for your letter and for the reviewers’ comments concerning our manuscript entitled “Morphological Changes and Components Characterization of Coffee Silverskin” (ID: molecules-1292519). Those comments are all valuable and very helpful for revising and improving our paper, as well as the important guiding significance to our researches. We have studied comments carefully and have made corrections which we hope meet with approval. Revised portions are marked in red on the paper. The main corrections in the paper and the responses to the reviewer’s comments are as flowing:
Point 1: p. 1 lines 29-30 “Effective use of these by-products is not only for environmental protection but also for environmental protection”, is not clear, please correct. 

Response 1: “NMR technology” has been replaced by “NMR spectroscopy” in the manuscript.
Point 2: Although the experimental section has been strongly revised, it should be further improved. Some paragraphs in section 3 “Materials and methods” are not appropriately described in my opinion. Specifically, some sentences should be rewritten in a more scientifically sound way, for instance at p 8 lines 206-209 “Take 1.5 kilograms of dried fruits, remove the pulp and parchment paper to obtain 760 grams of coffee beans containing CS, and divide them into three portions (A.350 g,B.350 g, C.60 g). […]” or lines 216-218 “Accurately weigh the CS powder of 0.5 g × 2 A sample and b sample and put them in 4 round-bottom flasks (5 mL), seal each sample and stir it with 3.5 mL D2O at 60 °C (1500 r/m) for 2h to obtain 4 water extracts”. Also, some sentences need revision to better clarify the meaning, i.e. p. 9 lines 236-240 “analyze and identify the 1D and 2D NMR spectra of the water extract of CS by combining the main chemical ingredients were identified against literature and the NMR spectrum of the mixed standard under the same conditions to ensure A(unroasted) and B (roasted) the accuracy of the component identification.”
Response 2: a). Lines 206-209 on page 8 were revised to “Dried fruits (1.5 kg) were removed the pulp and parchment to obtain 760 g of green beans covering with CS. Then, the beans were divided into three portions (A.350 g, B.350 g, C.60 g).”
b). Lines 216-218 on page 8 were revised to “Accurately weighed the CS powder of 0.5 g × 2 A sample and B sample and put them in 4 round-bottom flasks (5 mL), respectively; Sealed each sample and stired it with 3.5 mL D2O at 60 oC (1500 r/m) for 2 h to obtain 4 samples water extracts.”
c). Lines 236-240 on page 9 were revised to “The components of CS water extracts A (roasted) and B (unroasted) were analyzed and identified by using 1D NMR and 2D NMR spectra of related components [19-22], and the accuracy of component identification was confirmed by NMR spectra of the mixed standard samples under the same conditions.”
Point 3: Table S1. The spectral width in heteronuclear correlation experiments is given only for one dimension, please complete with missing information.
Response 3: The spectral width of the heteronuclear related experiment is 8012.82 Hz, which has been added to Table S1.
Reviewer 3 Report
The Authors have revised their manuscript and I acccept their explanations. This work can be published now.
Author Response
Dear Reviewers:
Thank you for your letter and for the reviewers’ comments concerning our manuscript entitled “Morphological Changes and Components Characterization of Coffee Silverskin” (ID: molecules-1292519). We appreciate Reviewers,warm work earnestly. Once again, thank you very much for your comments and suggestions
Reviewer 4 Report
Dear Authors, thank you for considering my comments and for updating your manuscript accordingly.
After reviewing the updated version, I only detected spelling problems in the literature section and tex. Especially in upper and lower case letters. Please take another round of editing (for example header 2.2.2 sugars, spelling of nmr, hplc, etc.).
Author Response
Dear Reviewers:
Thank you for your letter and for the reviewers’ comments concerning our manuscript entitled “Morphological Changes and Components Characterization of Coffee Silverskin” (ID: molecules-1292519). Those comments are all valuable and very helpful for revising and improving our paper, as well as the important guiding significance to our researches. We have studied comments carefully and have made corrections which we hope meet with approval. Revised portions are marked in red on the paper. The main corrections in the paper and the responses to the reviewer’s comments are as flowing:
Point 1: After reviewing the updated version, I only detected spelling problems in the literature section and tex. Especially in upper and lower case letters. Please take another round of editing (for example header 2.2.2 sugars, spelling of nmr, hplc, etc.). 

Response 1: The spelling problem in the manuscript has been resolved.